# Bone Morphogenetic Proteins for Nucleus Pulposus Regeneration

**DOI:** 10.3390/ijms21082720

**Published:** 2020-04-14

**Authors:** Anita Krouwels, Juvita D. Iljas, Angela H. M. Kragten, Wouter J. A. Dhert, F. Cumhur Öner, Marianna A. Tryfonidou, Laura B. Creemers

**Affiliations:** 1Department of Orthopedics, University Medical Center Utrecht, 3584 CX Utrecht, The Netherlands; 2Faculty of Veterinary Medicine, Utrecht University, 3584 CS Utrecht, The Netherlands; 3Department of Clinical Sciences, Faculty of Veterinary Medicine, Utrecht University, 3584 CS Utrecht, The Netherlands

**Keywords:** nucleus pulposus, regeneration, bone morphogenetic protein, heterodimer, proteoglycan, collagen

## Abstract

Matrix production by nucleus pulposus (NP) cells, the cells residing in the center of the intervertebral disc, can be stimulated by growth factors. Bone morphogenetic proteins (BMPs) hold great promise. Although BMP2 and BMP7 have been used most frequently, other BMPs have also shown potential for NP regeneration. Heterodimers may be more potent than single homodimers, but it is not known whether combinations of homodimers would perform equally well. In this study, we compared BMP2, BMP4, BMP6, and BMP7, their combinations and heterodimers, for regeneration by human NP cells. The BMPs investigated induced variable matrix deposition by NP cells. BMP4 was the most potent, both in the final neotissue glysosaminoglycan content and incorporation efficiency. Heterodimers BMP2/6H and BMP2/7H were more potent than their respective homodimer combinations, but not the BMP4/7H heterodimer. The current results indicate that BMP4 might have a high potential for regeneration of the intervertebral disc. Moreover, the added value of BMP heterodimers over their respective homodimer BMP combinations depends on the BMP combination applied.

## 1. Introduction

Low back pain, a considerable problem in today’s society [1], is associated with degeneration of the intervertebral disc (IVD) [2,3]. The first signs of IVD degeneration are visible in the nucleus pulposus (NP) [4], where loss of the glycosaminoglycan (GAG) content leads to the loss of water content impairing the biomechanical properties of the disc and eventually leading to further degeneration, pain, and decreased mobility in patients. Current treatments of chronic low back pain, conservative or surgical, do not address the underlying cause of degeneration. Growth factors are likely candidates to achieve biologic repair [5]. Among these growth factors are the bone morphogenetic proteins (BMPs) from the transforming growth factor-β (TGF-β) superfamily, which are commonly known for their involvement in bone formation. Since their discovery in the 1960s, they have been found to play important roles throughout the entire body, including formation and maintenance of cartilaginous tissues [6]. Within this context, the chondrogenic capacity of many BMPs is interesting for regeneration of the NP. BMP2 stimulates growth plate chondrocyte proliferation and hypertrophy to accelerate longitudinal bone growth [7], but was also shown to be important in postnatal cartilage development and maintenance [8]. BMP4 can induce and accelerate chondrogenic differentiation [9,10], while its co-expression with BMP2 is essential for osteogenesis [11]. BMP6 accelerates hypertrophic differentiation of chick chondrocytes [12,13]. BMP7 is a cartilage repair factor enhancing anabolic and reducing catabolic responses in cartilage and the IVD [14]. Noteworthy, BMP heterodimers were suggested to be more potent than homodimers, not only in bone formation [15,16,17,18,19], but also in NP regeneration [20]. However, it is not clear whether these effects require a heterodimeric conformation, or can be achieved by costimulation by combinations of the separate homodimers, as this was only once taken as a comparator, in ectopic bone formation [15].

For NP regeneration, mainly BMP2 and BMP7 have been investigated [21,22], most likely because they are clinically available [23]. In rabbit models of induced IVD degeneration, injection of BMP7 into the epidural space or the NPs relocated to the epidural space was shown to be safe [24], and dosages of 2 μg [25] or 100 μg [26] BMP7 per disc exerted regenerative effects. However, in a spontaneous degeneration model in beagle dogs, injection of dosages up to 250 μg per disc did not induce regeneration, but resulted in extensive extradiscal bone formation [27] indicating that human recombinant BMP7 was biologically active.

One reason why BMPs may fail to induce NP regeneration may reside in the limited number and/or capacity of degenerated NP cells to adequately respond to growth factor stimulation. NP cell collection from non-extruded discs for isolation and expansion may induce (further) disc degeneration, as nucleotomy is a well-known method of inducing degeneration [28] and hence is not considered a feasible clinical approach. To this end, mesenchymal stromal cells (MSCs) are being clinically employed for NP regeneration. The regenerative effects of MSCs are thought to be mediated by trophic effects and/or their differentiation toward NP cells [29,30,31,32,33]. To remedy the low amount of cells present, a combination of BMPs and MSCs could further augment IVD regeneration. In this scenario, BMPs are expected to both stimulate the resident NPs to proliferate and produce the matrix and facilitate MSCs to enter the chondrogenic lineage [34].

In the current study, we compared different BMP homodimers, heterodimers, and their respective homodimer combinations for their potency of inducing GAG formation by degenerated human NP cells in vitro. Furthermore, in a pilot experiment, we explored the effect of the two best performing BMPs, i.e., BMP4 and BMP4/7H, in co-cultures of NP cells and MSCs as the first step towards understanding how clinical applications of BMPs alone or in combination with MSC transplantation may augment NP regeneration.

## 2. Results

### 2.1. The Effects of BMPs on Degenerated Human NP Cells

#### 2.1.1. GAG Deposition

Degenerated human NP cells were cultured in high-density 2D cultures on type II collagen-coated transwell filters. In the preliminary experiment, the requirement of a growth factor for NP cell-mediated tissue formation was established by culturing in the presence or absence of TGF-β1 and TGF-β2 (see the Appendix A). The regenerative effects of BMP homodimers and heterodimers supplemented in equal concentration were compared by biochemical analysis of the GAG and DNA content (Figure 1). The cells cultured in the presence of BMP4 produced the highest amounts of GAGs, significant compared to BMP2 and BMP6 (*p* ≤ 0.001), but not BMP7. BMP2 induced the lowest amounts of GAG, GAG/DNA, and the total GAG content, defined as the GAG content of the construct plus the GAG released into the medium (*p* ≤ 0.005). Compared to the 0.4 nM TGF-β1 control, the cells cultured with BMP4 produced more GAG and GAG/DNA and had higher incorporation efficiency (*p* < 0.001). The cells cultured in the presence of BMP2 contained lower amounts of GAGs, DNA, and the total GAG than the cells with 0.4 nM TGF-β1 (*p* ≤ 0.01). Without a growth factor, a negligible amount of the matrix was produced (Appendix A).

Heterodimers BMP2/6H, BMP2/7H, and BMP4/7H at 4 nM were compared to their respective homodimer combinations cultured at 2 nM each to achieve a final concentration of 4 nM in order to determine whether heterodimers have a differential effect (Figure 1). Heterodimer BMP2/6H performed better in inducing GAG production (*p* = 0.002) than BMP2+6, but the GAG/DNA content did not differ between BMP2/6H and BMP2+6. The BMP2/7H treatment resulted in significantly higher GAG production and DNA content compared to BMP2+7 (*p* < 0.001). However, there were no differences at the GAG/DNA level. No differences were found between BMP4/7 H and BMP4+7 in GAG, DNA, GAG/DNA, or in the incorporation efficiency. Although heterodimers BMP2/6H and BMP2/7H induced higher GAG and DNA content and the total GAG than their homodimer combinations, GAG/DNA levels were not different. Any increase in matrix deposition thus seemed to have been caused by a higher cell number.

Complementary to the biochemical analysis, qualitative assessment of the deposited matrix was conducted. BMP treatment induced deposition of GAGs by the NP cells cultured on transwell filters, as indicated by positive Safranin O staining (Figure 1F). Overall, the proteoglycan-containing extracellular matrix was visible in the presence of any BMP as well as 0.4 nM TGF-β1, but the cells cultured in the presence of 4 nM TGF-β1 were packed together with very little extracellular matrix between them. The Safranin O-stained matrix cultured with BMP4, BMP2+6, BMP4+7, and heterodimers BMP2/6H, BMP2/7H, and BMP4/7H seemed to have a high intensity, whereas the matrix of the cells cultured with BMP2, BMP6, BMP7, BMP2+7, and 0.4 nM TGF-β1 stained less intense.

#### 2.1.2. Collagen Production and Deposition

In the same cultures, type II collagen production was also analyzed. Procollagen II (PIICP) as a measure of type II collagen production was higher in the NP cells treated with BMP4, BMP6, BMP7, and BMP2+7 than in the ones treated with BMP2/6H (*p* ≤ 0.037, Figure 2A). Collagen production by NP cells induced by BMP7 was also higher than the one induced by BMP2 and BMP4+7 (*p* ≤ 0.015), but no other differences were noted.

Type II collagen was also visualized with immunohistochemistry, mostly visible throughout the neotissue (Figure 2B), although only patches of positive staining were seen occasionally in the presence of 4 nM TGF-β1. There seemed to be no difference in type II collagen staining patterns between heterodimer BMP2/7H and combined homodimers BMP2+7 or between BMP4/7 H and BMP4+7, in line with the PIICP production. The type II collagen staining appeared more patchy with BMP2/6H compared with BMP2+6.

The presence of type I collagen in all the neotissue indicates that matrix formation was not exclusively chondrogenic (Figure 2C). Type I collagen was diffusely present throughout all the cultured neotissues, although the intensity appeared highest in the presence of BMP2+6 and lowest with BMP2/7H. Notably, less type I collagen seemed to be deposited in BMP2/7H compared to BMP2+7. BMP4/7H-treated constructs stained more intensely compared with BMP4+7-treated ones. In the presence of 4 nM TGF-β1, neotissue was primarily positive for type I collagen.

To check for possible hypertrophic differentiation, type X collagen immunohistochemistry was performed (Figure 2D). All the tissues were negative for type X collagen, while the vertebral growth plate of human fetal IVD tissue stained positive.

#### 2.1.3. Dose Dependency of BMP4 and BMP7 Stimulation

BMP4, BMP4+7, and BMP4/7H appeared to be particularly efficient in inducing GAG production by NP cells. To assess the effects of BMP4 in more detail and compare this to the more frequently used BMP7, dose dependency was determined with NP cells from 4 donors cultured on transwell inserts at concentrations between 0.04 and 4 nM (Figure 3). Any concentration of BMP4 induced more GAG production than the BMP-free control (*p* < 0.001), while 0.4, 2, and 4 nM BMP4 induced more GAG production than 0.04 nM BMP4 (*p* < 0.001). BMP7 induced more GAG production at 4 nM than the control, 0.04, and 0.4 nM (*p* ≤ 0.001) and 2 nM BMP7 (*p* = 0.009). The increased GAG production by BMP4 could partially be attributed to the increase in the DNA content (Figure 3B). Normalized per DNA, GAG deposition was highest for 0.4 nM BMP4 compared to the control, 0.04, and 4 nM BMP4 (*p* ≤ 0.005). In the presence of 4 nM BMP7, GAG/DNA was higher compared to all other concentrations (*p* ≤ 0.009). No differences in the total GAG production were measured between the BMPs or the concentrations used, but the variability was high. GAG incorporation was most efficient for 2 nM and 4 nM BMP4 compared to the control, 0.04, and 0.4 nM BMP4 (*p* ≤ 0.028). The highest GAG incorporation efficiency for BMP7-treated cells was measured in the presence of 4 nM BMP7 (*p* ≤ 0.003). Altogether, BMP4 was demonstrated to have an optimal effect on matrix production by the NP cells already around 0.4 to 2 nM (equivalent to 5.6 and 28 ng/mL), whereas BMP7 did not appear to reach its maximum effect until at 4 nM (equivalent to 63.2 ng/mL).

### 2.2. Pilot Study: The Effects of BMPs on Co-Cultures of NP Cells and MSCs

#### 2.2.1. MSC Multipotency

Multipotency assays for adipogenic and osteogenic differentiation of MSCs were performed in a 12-wells plate with an adipogenic or an osteogenic medium, and chondrogenic differentiation was achieved in pellet cultures with a chondrogenic medium. Adipogenic differentiation was confirmed by Oil Red O staining, positive staining of alkaline phosphatase (ALP) with fuchsine indicated osteogenic differentiation, and chondrogenesis was shown by deposition of GAGs in the extracellular matrix as indicated by positive Safranin O staining (Figure 4).

#### 2.2.2. BMPs with NP Cells Co-Cultured with MSCs

The MSC donor used in this pilot study did not enhance tissue formation by NP cells in the transwell culture system employed in the aforementioned studies, but actually seemed to inhibit GAG production by NP cells (data not shown). Therefore, NP cells were co-cultured with MSCs in a 50:50 ratio (often employed in literature [29,30,31,32,33,35]) in pellets in a differentiation medium only and in a differentiation medium supplemented with 0.4 nM TGF-β1 or BMPs. The BMP concentration was chosen based on the results of the dose dependency study depicted in Figure 3.

Co-culture of NP cells and MSCs in a 50:50 ratio in the presence of growth factors resulted in a significantly higher matrix deposition compared to the differentiation medium to which no additional growth factors were added. The GAG content was highest in the pellets cultured with BMP4 compared to the pellets cultred with BMP7, BMP4+7, and the differentiation medium with no added growth factors (*p* < 0.001, Figure 5A). The DNA content was higher for the pellets cultured with TGF-β1 compared to the pellets cultured with BMP7 and BMP4+7 (*p* ≤ 0.018, Figure 5B). GAG corrected for DNA was higher in the presence of all the growth factors compared to the differentiation medium (*p* < 0.001) and highest for BMP4 compared to TGF-β1, BMP7, BMP4/7H, and the differentiation medium (*p* ≤ 0.024, Figure 5C). Higher GAG/DNA was also measured in the pellets cultured with TGF-β1, BMP4+7, and BMP4/7H compared to the pellets cultured with BMP7 (*p* < 0.001). The total GAG production was higher for TGF-β1, BMP4, BMP4+7, and BMP4/7H compared to BMP7 and the differentiation medium (*p* < 0.001), and highest for TGF-β1 compared to BMP4, BMP7, BMP4+7, BMP4/7H, and the differentiation medium (*p* ≤ 0.033, Figure 5D). GAG incorporation efficiency was highest for BMP4 compared to TGF-β1, BMP7, and BMP4+7 (*p* ≤ 0.012). GAG, DNA, GAG/DNA, and the total GAG were higher for heterodimer BMP4/7H compared to the combination of homodimers (Figure 5E).

In TGF-β1, BMP4, or BMP7, the combination of homodimers, and the heterodimer, the co-cultured pellets showed positive staining for various matrix components in varying intensities, such as proteoglycans, indicated by Safranin O staining, and type II and type I collagen, indicated by immunohistochemistry (Figure 5F). The most intense Safranin O staining was observed in the pellets cultured with BMP4 followed by the combination of homodimers BMP4 and BMP7, while the most intense type II collagen staining was seen in the presence of BMP4 and heterodimer BMP4/7H. In the presence of BMP4+7, the pellets produced only limited type II collagen. The pellets cultured with TGF-β1 showed the most intense type I collagen staining, with only limited Safranin O and type II collagen staining.

## 3. Discussion

In the current study, we have shown that BMPs, heterodimers, and their homodimer combinations induced proteoglycan formation and deposition, and deposition of type I and type II collagen by degenerated human NP cells. In particular, the NP cells cultured with equal concentrations of BMP4, BMP4+7, BMP6, and heterodimers BMP2/7H and BMP4/7H showed more extensive matrix deposition by NP cells compared to the NP cells cultured with BMP2 and its combinations, BMP7, or TGF-β1. The differences in matrix deposition could partly be attributed to an increase in the cell number. Furthermore, GAG incorporation was most efficient for the NP cell cultures treated with BMP4+7, BMP2/7H, and BMP4/7H. Altogether, these findings indicate that heterodimers of BMPs can display enhanced regenerative effects compared to their respective combinations of homodimers, but the effects were frequently not different from their most potent homodimers alone or the combination of the respective homodimers.

Among the different BMP homodimers tested in the present study, BMP4 outperformed the other BMPs. Although BMP6 also induced high GAG production, it was significantly lower than when induced by BMP4, especially when looking at the total production over time. Furthermore, when the commonly used BMP7 was compared to BMP4 in different concentrations, BMP4 induced significantly more matrix formation by NP cells at much lower concentrations. In co-cultures of NP cells and MSCs of one donor, most regeneration was accomplished by the addition of BMP4 rather than a heterodimer. The present study shows that of the BMPs tested here, BMP4 may be the most suitable candidate for NP regeneration, although BMP2 and BMP7 thus far have been the usual candidates employed in regenerative treatment strategies of the degenerated disc [21,22,24,25,26,27,36,37,38]. In line with these findings, induction of chondrogenesis in MSCs has been shown to be more efficient with BMP4 than with BMP7, and these cells produced a better proteoglycan-rich matrix in vivo [39]. Our results seem to contrast with a previous study in which bovine NP cells in the monolayer were transduced with vectors for various BMPs (BMP2, -3, -4, -5, -7, -8, -10, -11, -12, -13, -14, and -15 and SOX9) [40]. Here, BMP2 and -7 were the most effective in inducing GAG production, followed by BMP4. BMP4 and -14 induced the highest type II collagen content. However, the physiology of healthy bovine NP cells does not necessarily resemble human NP cells from degenerated discs [41]. It also remains to be determined whether concentrations of the various BMPs produced by the transfected bovine NP cells were equal and hence can provide solid evidence on the differences in biologic potency of the BMPs studied. The difference in effects between the BMPs found here may be explained by the fact that they operate via different pathways, as has been shown for BMP2 and BMP7 in human NP cells [38] and BMP4 and BMP7 in human MSCs [42]. Signaling of various BMPs in development is also different across different joints in mice at birth [11]. BMP4 seemed to exert its effects partially by enhancing the cell content, but also by enhancing synthetic activity of the NP cells. Altogether, these findings indicate that BMP4 may be a better candidate for future NP regenerative strategies.

Notably, while BMP4/7H compared to homodimer combination BMP4+7 did not show stronger induction of extracellular matrix production on degenerated NP cells in a monoculture, it did exhibit an additive anabolic and proliferative effect on the NP cells co-cultured with MSCs in a pilot study, as shown previously for bone regeneration [15]. This difference might be attributed to cell type-specific signaling, although the mechanism behind possible enhanced potency of heterodimers is not clear. The potency of heterodimers was demonstrated in bone formation in Chinese hamster ovary (CHO) cells, rats, mini-pigs, and goats [16,17,18], human embryonic stem cell differentiation [43], and murine blood plasma [44], although here the combination of separate homodimers was not included. In cartilaginous tissues, the effects of heterodimers have been studied to a limited extent only [20,45], also omitting the comparison to (a combination of) homodimers. How heterodimers may exert differential effects over (combined) homodimers is not completely clear. Heterodimers were shown to have different receptor binding properties than homodimers, and thereby enhance SMAD signaling more than the homodimer receptor binding [46,47], also compared to combined homodimers [48], and might have reduced affinity for antagonists compared to homodimers [48,49]. The stability of the BMP-to-receptor-binding also seems to be higher for heterodimers compared to homodimers, which may provide another explanation [46]. Moreover, in *Drosophila*, enhanced receptor binding was found to give differential effects that could not be explained by altered downstream signaling alone [50]. As such, the present study provides clear evidence that heterodimers can be more potent than the respective homodimer combinations, both in a culture with NP cells only and in co-cultures of NP cells with MSCs, although this was not the case for all combinations.

TGF-β is frequently used in cultures of NP cells, because without growth factor stimulation, they produce little matrix as has been shown previously [51,52]. It does, however, also have unwanted effects, such as increasing expression of type I collagen and MMP13 [53]. Interestingly, differentiation of MSCs is enhanced by TGF-β [54], but gene expression patterns for NP-specific matrix components in human cervical and lumbar NP cells were more NP-like with BMP2 than with TGF-β [52]. This might account for the high amounts of GAGs produced in the co-cultures in the presence of TGF-β in the current study and also for the fact that in NP cells, BMPs were more efficient in inducing GAG production. The matrix produced in the presence of TGF-β appeared also of lower quality than the matrix produced in the presence of BMP4 as indicated by the lower incorporation efficiency and the more intense type I collagen staining.

MSCs are currently being used to prevent NP degeneration or enhance NP regeneration in vitro [29,30,31,32,33,55] and in animal models [31,56,57] and have been shown to be safe in the clinical trials reported [58,59] and ongoing (for example, NCT01290367, NCT01860417, NCT02338271). Co-cultures of MSCs and NP cells have been reported to enhance matrix production by (degenerated) NP cells [54,60,61]. Although differentiation of MSCs toward an NP-like phenotype has been shown [60], enhanced matrix production is thought to be due to the trophic effects MSCs have on other cells [62,63]. In the current study, when growth factors, especially BMP4, were added to the culture medium of co-cultures, matrix production seemed to be enhanced. BMPs are expensive products and limited in use due to short half-lives [64], MSCs could be employed as carriers for BMPs in addition to their proteoglycan production. Possibly, BMP production by transfected MSCs can induce regeneration by resident NP cells and implantation of these genetically modified MSCs might be a very interesting treatment strategy [65,66,67,68,69,70,71].

In conclusion, although BMP2 and BMP7 have been clinically approved, they are not the most potent inducers of regeneration of degenerated human nucleus pulposus cells. Instead, BMP4 emerges from this study as a potent inducer of regeneration by NP cells. It remains to be determined if this also applies to the NP cells combined with MSCs. Heterodimers were found in some cases to be more potent than the combination of homodimers, which showed a cell- or culture-system dependency, and hence heterodimers should always be compared to a combination of the respective homodimers.

## 4. Materials and Methods

### 4.1. Cultures

#### 4.1.1. Cell Isolation

Intact lumbar IVDs of 11 donors (age 47 to 68, mean age 53) were obtained as part of the standard postmortem procedure as approved by the medical ethics committee of the University Medical Center Utrecht (METC No. 12-364). The vertebrae were removed along the endplates and the tissue was graded for degeneration according to the Thompson classification [72]. Only Thompson grade III IVDs were used in this study. NP tissue was conservatively separated from annulus fibrosus tissue, and cells were isolated using 1 h incubation with 14 U/mL Pronase (Roche, Basel, Switzerland) and 110 U/mL deoxyribonuclease (DNAse) II (Sigma, St. Louis, MO, USA), followed by overnight incubation with 13.7 U/mL collagenase type 2 (Worthington, Lakewood, NJ, USA) and 110 U/mL DNAse, all diluted in a Dulbecco’s modified Eagle’s medium (DMEM, low-glucose, pyruvate, L-glutamine, 31885, Gibco, Carlsbad, CA, USA). NP cells were then plated and expanded with DMEM + 10% heat-inactivated fetal bovine serum (FBS, HyClone, Logan, UT, USA) + 200 U/mL penicillin, 200 µg/mL streptomycin, 50 mg/L amphotericin-B, and 50 mg/L gentamycin (Lonza, Basel, Switzerland) and 10 ng/mL recombinant human basic fibroblast growth factor (bFGF, with BSA as the carrier, R&D Systems, Minneapolis, MA, USA) for 2 passages before use in regeneration cultures. MSCs were isolated from bone marrow of a 69-year old female undergoing arthroplasty (METC number 08-001) by centrifugation on Ficoll-Paque PLUS (GE Healthcare, Little Chalfont, UK) and expanded in alpha-minimum essential medium (α-MEM, Gibco) + 10% FBS + 1% ascorbic acid-2-phosphate (ASAP, Sigma) + 100 U/mL penicillin + 100 µg/mL streptomycin (pen/strep, Gibco) + 1 ng/mL bFGF. After three days, the MSC cultures were washed to remove non-adherent cells.

#### 4.1.2. BMPs in NP Differentiation Culture

For differentiation, NP cells of 4 donors were seeded in high density on 0.12 cm^2^ type II collagen-coated polycarbonate film (PCF) 0.4 µm transwell filters (Merck Millipore, Billerica, MA, USA) at passage 2 at 1*10^6^ cells/cm^2^ with a differentiation culture medium (DMEM + 2% insulin-transferrin-selenium-X (ITS-X, Gibco) + 0.4 nM ASAP + 2% human serum albumin (Sanquin, Amsterdam, the Netherlands) + pen/strep. BMP2, BMP4, BMP6, BMP7, heterodimers BMP2/6H, BMP2/7H, and BMP4/7H (all with BSA as the carrier, R&D Systems) were added to the culture medium at 4 nM, and combinations of BMP2+6, BMP2+7, and BMP4+7 homodimers were added at 2 nM per homodimer. As the positive control, 0.4 and 4 nM TGF-β1 (10 and 100 ng/mL, respectively, R&D Systems) were included based on the fact that TGF-β1 and TGF-β2 retrieved similar anabolic effects on human NP cells in the same culture setup (Appendix A). Based on the matrix production in the comparison between the different BMP homodimers tested in the present study, BMP4 and BMP7 were also applied to 4 donors at the concentration of 0.04, 0.4, 2, and 4 nM. Cultures were performed for 28 days in a humidified incubator at 37 °C and 5% CO_2_. The medium was changed 3 times per week and stored at −80 °C until further analysis. The culture medium was collected during the entire culture period and stored at −80 °C for GAG and PIICP analysis.

#### 4.1.3. MSC Multipotency Assays

Multipotency of isolated MSCs was assessed by their capacity to differentiate towards an adipogenic, osteogenic, and chondrogenic lineage [73]. For adipogenic and osteogenic differentiation, the cells were plated in 12-well plates at 6000 cells/cm^2^ and grown until confluency. Then, an adipogenic medium (α-MEM + 10% heat-inactivated FBS + pen/strep + 1 µM dexamethasone (Sigma, St. Louis, MO, USA) + 0.5 mM IBMX (3-isobutyl-1-methylxanthine, Sigma) + 0.2 mM indomethacin (Sigma) + 1.72 µM insulin (Sigma)) or an osteogenic medium (α-MEM + 10% FBS + 0.2 nM ASAP + pen/strep + 10 mM β-glycerophosphate (Sigma) + 10 nM dexamethasone (Sigma)) was added to the cultures. 250,000 cells were pelleted and cultured with a chondrogenic medium (DMEM + 0.2 nM ASAP + 1% ITS-Premix (Fisher Scientific, Waltham, MA, USA) + pen/strep + 0.1 µM dexamethasone + 0.4 nM TGF-β1, with BSA as the carrier, R&D Systems). After 21 days, Oil Red O staining was performed to assess adipogenesis. After 7 days, alkaline phosphatase (ALP) produced by the osteogenically differentiated MSCs was visualized with fuchsinw (Fuchsin + kit, K0625, DAKO, Glostrup, Denmark). Chondrogenically differentiated pellets were fixed with 4% buffered formalin, embedded in paraffin, and Safranin O/Fast Green/hematoxylin staining was performed as described below.

#### 4.1.4. MSC and NP Cell Co-Culture

As MSCs did not thrive in the transwell culture system, co-cultures were performed in a pellet culture. Since no differences were found between NP cell ratios (data not shown), subsequent co-culture experiments with BMPs were done with the NP:MSC cell pellets with a ratio of 50:50, which are also frequently used in literature [29,30,31,32,33,35]. Pellets of a single MSC donor (at passage 4) with 4 NP cell donors were formed by centrifugation of U-bottom 96-well plates containing cell suspensions in a differentiation culture medium at 1750 rpm (616× *g*) for 2 min. The medium was changed every 3 days. Pellets were formed with MSCs and NP cells (50:50) and cultured in the presence of 0.4 nM BMP4, BMP7, BMP4+7 (0.2 nM each), BMP4/7H, or TGF-β1.

### 4.2. Analyses

#### 4.2.1. Biochemistry: Tissue GAG and DNA Content

From 3 to 5 cultured constructs or pellets per donor and treatment group were digested in a digestion buffer (250 μg/mL papain and 1.57 mg/mL L-cysteine (Sigma)). A PicoGreen assay (Life Technologies, Carlsbad, CA, USA) was used to determine the DNA content against a λ-DNA standard curve. Fluorescence was measured in a POLARstar Optima fluorescence microplate reader (Isogen Life Science, Utrecht, The Netherlands) at 485 nm excitation and 530 nm emission. Glycosaminoglycan (GAG) content of the digested samples or the medium was measured with a 1,9-dimethylmethylene blue (DMMB, Sigma) assay and quantified against a chondroitin-6-sulfate standard curve. The optical density (OD) was determined at 525 nm and reference wavelength 595 nm with a VersaMax microplate reader and the SoftMax Pro software (Molecular Devices, San Jose, CA, USA).

#### 4.2.2. Procollagen II ELISA

The carboxy-terminal propeptide of the type II collagen (PIICP) content in the culture medium is directly related to type II collagen production during the culture period [74]. We have previously shown that most PIICP is produced in week 2 and is correlated to the total amount formed during the culture period [75]. The PIICP was measured on 3 medium samples per donor and treatment group using an enzyme-linked immunosorbent assay (ELISA, Cloud-Clone Corp., Houston, TX). The samples and the standard diluted in the culture medium were incubated for 2 h, after which incubations with detection antibodies were performed for 1 h and, after washing, 30 min subsequently. Then the plate was washed and incubated with a 3,3′,5,5′-tetramethylbenzidine (TMB) substrate solution for 20–25 min. After addition of the stop solution, the plate was measured at 450 nm on a VersaMax microplate reader.

#### 4.2.3. Histology

Two cultured constructs or pellets per donor and treatment group were fixed for two days with 10% buffered formalin (J.T.Baker, Avantor Performance Materials, Center Valley, PA, USA), dehydrated though a graded series of alcohol and embedded in paraffin. Sections of 5 µm were stained with 0.125% Safranin O for GAGs and 0.4% Fast Green for collagens.

#### 4.2.4. Immunohistochemistry

Immunohistochemistry for aggrecan, type I, type II, and type X collagen were performed after 0.03% hydrogen peroxidase activity blocking and antigen retrieval with Pronase (1 mg/mL, Sigma) and subsequent hyaluronidase (10 mg/mL, Sigma) incubation (for type I and II collagen) or cooking in a citrate buffer (10 mM sodium citrate, pH 6.0, for aggrecan). Antigen retrieval for type X collagen was performed with pepsin (1 mg/mL, Sigma) incubation for 2 h at pH 2.0 followed by hyaluronidase at 37 °C. After antigen retrieval, protocols were the same for all antibodies. Primary antibodies for type I collagen (2 µg/mL, rabbit, EPR7785, Abcam, Cambridge, UK), type II collagen (0.4 μg/mL, mouse, DSHB II-II6B3, DSHB, Iowa City, IA, USA), type X collagen (10 μg/mL, mouse X53, Quartett, Berlin, Germany), or aggrecan (1:150, mouse, 6-B-4, Abcam), or isotype controls (DAKO, concentrations matched with the primary antibody), were diluted in PBS + 5% BSA and incubated overnight at 4 °C, after which species-specific Envision kits (DAKO) and 3,3′-diaminobenzidine tetrahydrochloride hydrate were used for visualization. Sections were counterstained with the Mayer’s hematoxylin solution and dehydrated and mounted with DePeX. Microscopy pictures were taken with an Olympus BX51 upright microscope and the Cell^F software (Olympus, Tokyo, Japan).

### 4.3. Statistics

Statistical analyses were performed using the SPSS software (IBM, Armonk, NY, USA). For the normally distributed procollagen ELISA data, a univariate ANOVA with randomized block design with the post hoc Tukey *t*-test was performed. Otherwise, the non-parametric Kruskal–Wallis test with post hoc multiple Mann–Whitney comparisons were performed with the Benjamini–Hochberg correction for multiple testing. Outliers detected with the five-sigma rule (values outside the mean ± 5 * standard deviation range) were removed. The data are displayed as the mean ± the SD.

## Figures and Tables

**Figure 1 ijms-21-02720-f001:**
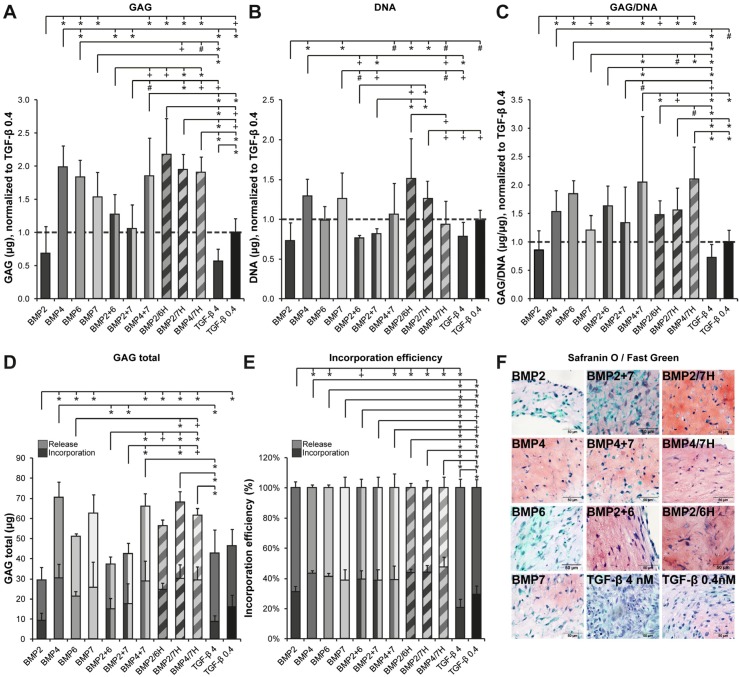
Glycosaminoglycan (GAG) production by human degenerated nucleus pulposus (NP) cells from 4 donors cultured on filters coated with type II collagen and treated with different bone morphogenetic proteins (BMPs, 4 nM total concentration, also with combinations of homodimers and heterodimers (H)) and compared to transforming growth factor (TGF)-β1 controls (4 and 0.4 nM, equal to 100 and 10 ng/mL). GAG and DNA content, GAG/DNA are displayed relative to 0.4 nM TGF-β1-treated NP cells, which were set at 1 (indicated by the dashed line). All the data are *n* = 12 (3 separate repetitions per donor per condition). (**A**) GAG content was highest in differentiation cultures with BMP4 and BMP2/6H and lowest for 4 nM TGF-β1 and BMP2. (**B**) DNA content was highest when cells were cultured with BMP2/6H, BMP4, BMP2/7H, and BMP7. (**C**) The amount of GAG corrected for DNA was highest in BMP4/7H, followed by BMP4+7, BMP6, and BMP4. (**D**) Total GAG production, including the GAGs released into the medium and contained in the neotissue, was highest in BMP4, BMP2/7H, and BMP4+7. (**E**) Incorporation efficiency was highest in the cells cultured with BMP4/7H. (**F**) Safranin O/Fast Green staining on histological sections of the NP cells cultured on transwell filters reveals that GAGs were deposited in the presence of all BMPs, although the staining was less intense for BMP2, BMP6, or BMP7. Cells in the presence of controls with 0.4 and especially 4 nM TGF-β1 produced hypercellular tissues with limited extracellular matrix (ECM). Representative images for all donors are shown. Significant differences are indicated as follows: * *p* ≤ 0.001, + *p* ≤ 0.005, # *p* ≤ 0.01, ^ *p* ≤ 0.025, $ *p* ≤ 0.05.

**Figure 2 ijms-21-02720-f002:**
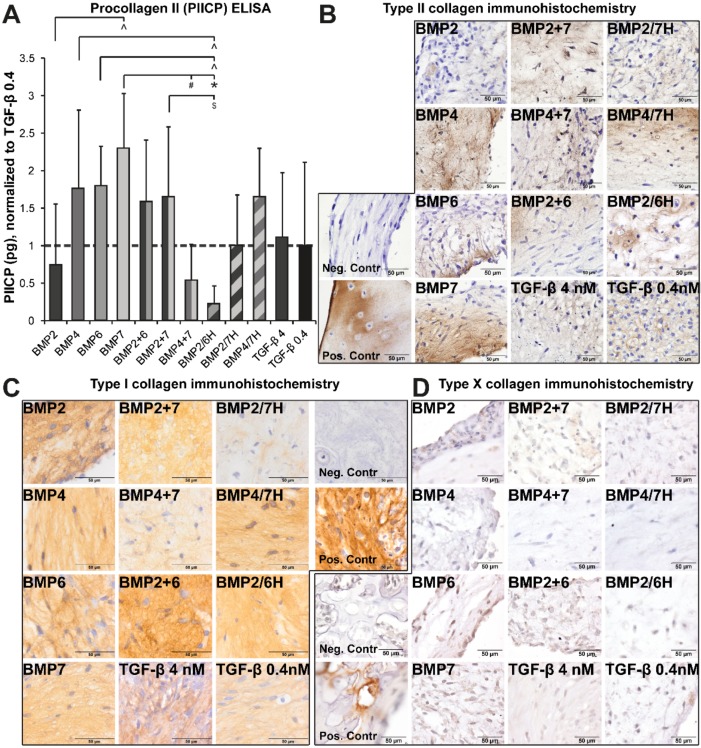
Human degenerated nucleus pulposus (NP) cells were cultured on type II collagen-coated transwell inserts in the presence of 4 nM bone morphogenetic proteins (BMPs, alone, in combination, or heterodimers (H)) or 0.4 or 4 nM transforming growth factor (TGF)-β1. (**A**) Procollagen type II production (PIICP), measured by enzyme-linked immunosorbent assay (ELISA), is displayed relative to 0.4 nM TGF-β1-treated NP cells, which were set at 1 (indicated by the dashed line). The data are *n* = 12 (3 separate repetitions per donor per condition). Procollagen II production in week 2 was higher by the NP cells cultured with BMP7 than with BMP2, BMP4+7, and BMP2/6H. The cells cultured with BMP4, BMP6, and BMP4+7 also produced more type II procollagen than the cells cultured with BMP2/6H. Significant differences are indicated as follows: * *p* ≤ 0.001, + *p* ≤ 0.005, # *p* ≤ 0.01, ^ *p* ≤ 0.025, $ *p* ≤ 0.05. (**B**) Immunohistochemistry shows that type II collagen was deposited in all the neotissues, but only localized staining was seen occasionally in the presence of 4 nM TGF-β1. The negative control is a cultured construct, the positive control is an osteoarthritic cartilage. (**C**) All the constructs cultured with BMPs and TGF-β1 were positive for type I collagen visualized by immunohistochemistry. The intensity was higher for BMP2+6 and very faint for BMP2/7H. Both negative and positive controls are cultured constructs. (**D**) Type X collagen was not detectable in any of the neotissues while the growth plate was immunopositive. Both negative and positive controls are fetal vertebra.

**Figure 3 ijms-21-02720-f003:**
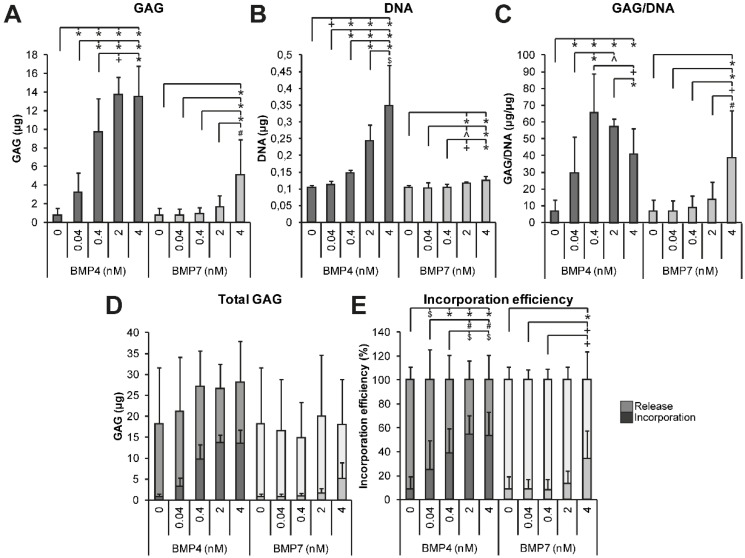
Dose response of human degenerated nucleus pulposus (NP) cells of 4 donors (*n* = 12; 3 separate repetitions per donor per condition) to different concentrations of bone morphogenetic proteins (BMP)4 and BMP7 was measured at the biochemical level. (**A**) Glycosaminoglycan (GAG) production increased with increasing BMP4 or BMP7 concentration. (**B**) Higher DNA content was measured with increasing BMP4 and BMP7 concentration. The effect of BMP7 was less pronounced than of BMP4. (**C**) GAG/DNA was highest for 0.4 nM BMP4. In the presence of 4 nM BMP7, GAG/DNA was higher than with the control or lower concentrations of BMP7. (**D**) No differences were measured for the total amount of GAG, including the GAGs released into the medium. (**E**) Incorporation efficiency was highest for 2 nM BMP4 and 4 nM BMP4. Higher incorporation efficiency was measured in the presence of 4 nM BMP7 compared to the control and lower concentrations of BMP7. Significant differences are given as follows: * *p* ≤ 0.001, + *p* ≤ 0.005, # *p* ≤ 0.01, ^ *p* ≤ 0.025, $ *p* ≤ 0.05.

**Figure 4 ijms-21-02720-f004:**
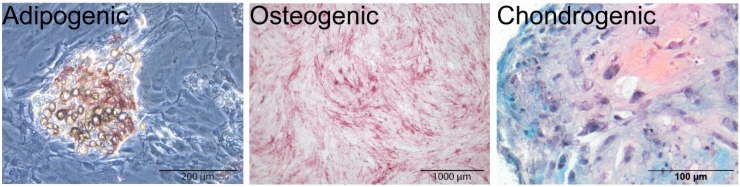
Multipotency assays were performed to ascertain the tri-potency of the human mesenchymal stromal cells (MSCs) used in this study. Red staining by Oil Red O indicated lipid formation after adipogenic differentiation of MSCs. Alkaline phosphatase was stained in pink by fuchsine, indicating that osteogenic differentiation occurred. Chondrogenesis was confirmed by Safranin O staining in pellets.

**Figure 5 ijms-21-02720-f005:**
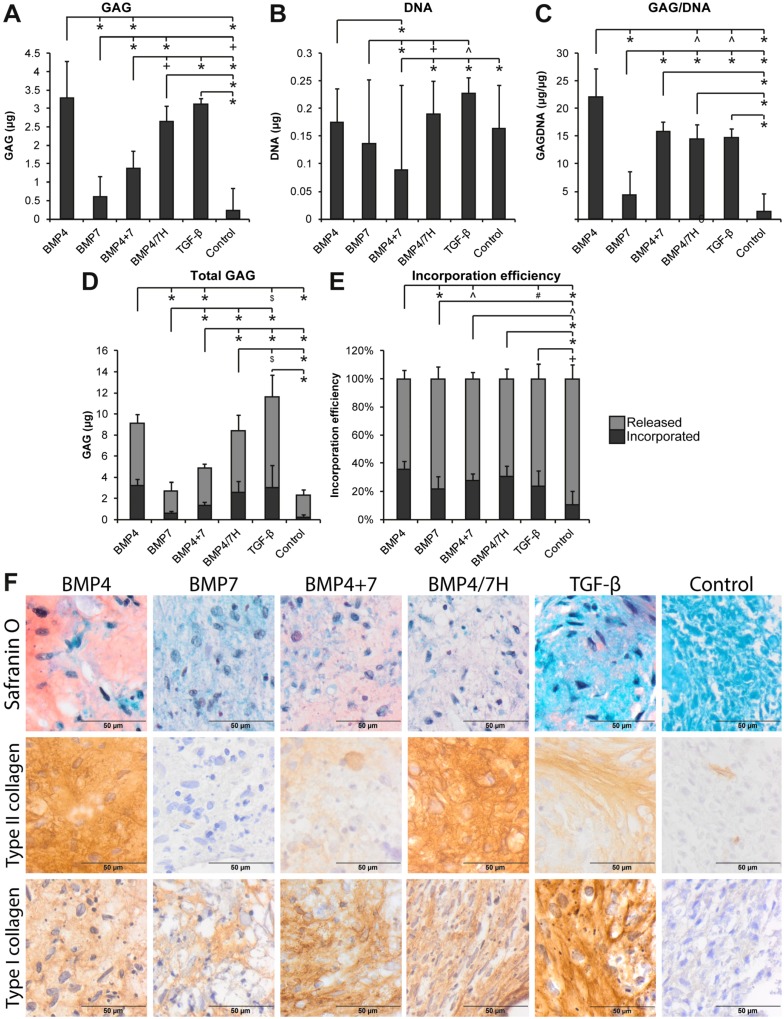
Matrix production and deposition of 50:50 nucleus pulposus: mesenchymal stromal cells (NP:MSC) pellet co-cultures in the presence of 0.4 nM bone morphogenetic proteins (BMPs), combination of homodimers, or heterodimer (H) were compared to 0.4 nM transforming growth factor (TGF-β1) or the control (differentiation culture medium without BMPs or TGF-β1). NP cells from 3 donors were co-cultured with MSCs from a single donor. All the data are *n* = 12 (4 separate repetitions per NP cell donor per condition). (**A**) Glycosaminoglycan (GAG) content was highest for BMP4, TGF-β1, and BMP4/7H. (**B**) DNA content was higher for the pellets cultured with TGF-β1 compared to BMP7 and BMP4+7. (**C**) GAG corrected for DNA was higher for all the growth factors compared to the control. (**D**) The total GAG production, determined as the GAGs in the tissue as well as released into the medium, was higher for TGF-β1, BMP4, BMP4+7, and BMP4/7H compared to BMP7 and the control. (**E**) Incorporation efficiency calculated as the percentage of the GAG incorporated into the tissue of the total GAG, was highest for BMP4. Significant differences are given as follows: * *p* ≤ 0.001, + *p* ≤ 0.005, # *p* ≤ 0.01, ^ *p* ≤ 0.025, $ *p* ≤ 0.05. (**F**) Safranin O staining was present in all the growth factor-treated MSC+NP pellets, but not in controls. The most intense staining was seen in the pellets cultured with BMP4. Type II collagen immunopositivity was present in all the pellets except for controls. The most intense staining was seen in BMP4- and BMP4/7H-treated pellets. The most intense type I collagen staining was visible in the pellets cultured with TGF-β1; it was also present when cultured with the various BMPs. Only control pellets were negative for type I collagen.

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
