# Peer review of "Bone Morphogenetic Proteins for Nucleus Pulposus Regeneration"

_ijms, 2020, doi:10.3390/ijms21082720_

Round 1
Reviewer 1 Report
This is a nice and very thorough performed study.
There are very few minor comments to further improve the manuscript
or increase comprehensiveness:
Page 3, line 105, (Figure 1F - here follows an error message:
Error! Reference source not found) Please check!
Are the results shown in Figure 1 the results from 4 Independent
repetitions with cells from 4 donors - or did the authors use a pool
of cells from the 4 donors - and the graph shows the mean of n=3 per condition?
This is not clear from the figure legend.
In Figure 2b two IHC controls are shown which look like tissue sections.
Please add in the legend the missing Information regarding these controls.
Page 7 of 18, Section 2.2.2 BMPs with NP cells cocultured with MSCs
Line 206/207: ….but actually seemed to inhibit production by NP cells.
"Production" of what? Please add missing Information!
Discussion Section, page 9 of 18, line 291,
Here the authors mention "regenerative" effects.
The term "regenerative" is a bit too ambitious as monolayer cultures of NP cells do not
really represent a model for regeneration. Please describe more precisely the respective
effect.
Author Response
Response to reviewer 1 comments
Point 1: Page 3, line 105, (Figure 1F - here follows an error message:
Error! Reference source not found) Please check!
Response 1: Apparently, the cross-referencing to figures did not transfer well, it is now replaced by regular text. Any other cross-references were removed as well.
Point 2: Are the results shown in Figure 1 the results from 4 Independent
repetitions with cells from 4 donors - or did the authors use a pool
of cells from the 4 donors - and the graph shows the mean of n=3 per condition?
This is not clear from the figure legend.
Response 2: The results shown in figure 1 were from independent repetitions with cells of 4 donors (n=3 for each condition for each donor). The legends of all figures where this is relevant have been amended.
Point 3: In Figure 2b two IHC controls are shown which look like tissue sections.
Please add in the legend the missing Information regarding these controls.
Response 3: The positive control for type II collagen is a section of osteoarthitic cartilage, both controls for type X collagen are sections of a fetal vertebral body. The missing information has been added to the legend.
Point 4: Page 7 of 18, Section 2.2.2 BMPs with NP cells cocultured with MSCs
Line 206/207: ….but actually seemed to inhibit production by NP cells.
"Production" of what? Please add missing Information!
Response 4: The production of GAGs seem to be inhibited when compared to monocultures of NP cells. The word “GAG” has been added to the sentence.
Point 5: Discussion Section, page 9 of 18, line 291,
Here the authors mention "regenerative" effects.
The term "regenerative" is a bit too ambitious as monolayer cultures of NP cells do not
really represent a model for regeneration. Please describe more precisely the respective
effect.
Response 5: The word “regenerative” is replaced by “induction of extracellular matrix production”
Reviewer 2 Report
Very interesting paper on the ability of different BMP family members to induce matrix deposition by NP cells and MSC.
The findings shown here are of value for both fundamental and applied BMP biology given the potential in intervertebral disk degeneration treatments.
The research design is very well thought and the care taken by the authors to test single BMPs, heterodimers on mixture of corresponding BMPs give a wealth of informations.
In consequence I fully support the publication of this study.
Author Response
Thank you